# Korean physicians' perceptions regarding disclosure of patient safety incidents: A cross-sectional study

Jeehee Pyo[1,2], Eun Young Choi[1,3], Won Lee[4], Seung Gyeong Jang[5], Young-Kwon Park [6], Minsu Ock [1,6,7]*, Sang-Il Lee[7]

1 Department of Preventive Medicine, Ulsan University Hospital, University of Ulsan College of Medicine, Ulsan, Republic of Korea, 2 Asan Medical Institute of Convergence Science and Technology, Asan Medical Center, University of Ulsan College of Medicine, Seoul, Republic of Korea, 3 Department of Nursing, Graduate School of Chung-Ang University, Seoul, Republic of Korea, 4 Department of Nursing, Chung-Ang University, Seoul, Republic of Korea, 5 College of Nursing, The Catholic University of Korea, Seoul, Republic of Korea, 6 Prevention and Care Center, Ulsan University Hospital, Ulsan, Republic of Korea, 7 Department of Preventive Medicine, University of Ulsan College of Medicine, Seoul, Republic of Korea

* ohohoms@naver.com

**Data Availability Statement:** All relevant data are within the manuscript.

**Funding:** This work was supported by a National Research Foundation of Korea (NRF) grant, funded

## Abstract

The present study investigated physicians' perceptions regarding the need for, effects of, and barriers to disclosure of patient safety incidents (DPSI). An anonymous online questionnaire survey was conducted to investigate physicians' perception regarding DPSI, in particular of when DPSI was needed in various situations and of methods for facilitating DPSI. Physicians' perceptions were then compared to the general public's perceptions regarding DPSI identified in a previous study. A total of 910 physicians participated. Most participants (94.9%) agreed that any serious medical error should be disclosed to patients and their caregivers, whereas only 39.8% agreed that even near-miss errors, which did not cause harm to patients, should be disclosed. Among the six known effects of DPSI presented, participating physicians showed the highest level of agreement (89.6%) that "DPSI will lead physicians to pay more attention to patient safety in the future." Among six barriers to DPSI, participants showed the most agreement (75.9%) that "It is unreasonable to demand DPSI in only the medical field, and disclosure is not actively conducted in other fields." With respect to methods for facilitating DPSI, participants agreed that "A guideline for DPSI is needed" (91.2%) and "Manpower to support DPSI in hospitals is required" (89.1%). Meanwhile, 79.3% agreed that "If an apology law is enacted, physicians will perform more DPSI" and 72.4% that "I support the introduction of an apology law." Korean physicians generally have a positive perception of DPSI, but less than the general public.

## Introduction

When a patient safety incident occurs, it is important for medical professionals to handle it well in terms of quality and ethics of care [1]. Systematic and effective measures in response to such patient safety incidents are ensured by disclosure of patient safety incidents (DPSI), defined as follows [2]: "When a patient safety incident occurs, medical professionals preemptively explain

by the Korea government (MSIT; No.
2018R1C1B6005186). The funder had no role in
the study design, data collection and analysis,
decision to publish, or preparation of the
manuscript.

**Competing interests:** The authors have no
potential conflicts of interest.

the incident to the patients and their caregivers, express sympathy and regret for the incident, deliver an apology and compensation appropriately if needed, and promise to prevent recurrence." The emphasis on DPSI in the field of patient safety is based both on ethical considerations [1, 3] and on various known beneficial effects including reduced number of medical lawsuits and related costs; reduced intention to discipline medical staff; stronger physician–patient relationships; increased intention to revisit and recommend medical staff; higher assessment scores for quality of care; and reduced guilt among medical staff [4].

However, it is known to be difficult for medical staff to perform DPSI in actual clinical practice [2, 5]. Medical staff are reluctant to perform DPSI even under situations when DPSI is needed due to fears about medical lawsuits and disciplinary actions, as well as about losing the trust of patients and colleagues [4]. In particular, among physicians, who are in the most critical position with regard to performing DPSI, not having accurate perceptions of DPSI and doubting its effects and benefits act as a major barrier to proper execution of DPSI [2]. It is known that physicians, who are often in a position to lead and supervise the entire patient treatment process, are highly likely to proactively decide how to handle patient safety incidents, and that patients and their caregivers want to hear an explanation about the patient safety incident from physicians [2].

Therefore, to improve DPSI in clinical practice, it is necessary to improve physicians' perceptions regarding DPSI, which would first require assessment of how physicians perceive DPSI. While there are previous studies quantitatively examining physicians' perception regarding DPSI [6–11], most of these are not recent studies and thus do not reflect the latest trends. Moreover, because most of these studies were conducted in Western countries, possible cultural differences in perception regarding DPSI cannot be dismissed. In particular, very few studies quantitatively examine the differences in perceptions of DPSI between physicians and the general public, including patients [6–8]. Considering that narrowing the differences in opinions on DPSI between physicians and the general public, including patients, is likely the first step toward activating DPSI in clinical practice, it seems necessary to clearly ascertain differences in perception of DPSI between physicians and the general public (henceforth, including patients) [10].

Accordingly, the present study conducted a questionnaire survey among physicians in Korea to investigate their perceptions of DPSI from various perspectives, including its effects, barriers to it, and methods facilitating it. In addition, the findings were compared to results obtained from a questionnaire survey about DPSI conducted on the general public to examine differences in perceptions.

## Materials and methods

The present study focused on the results obtained from a questionnaire survey of physicians within a bigger project investigating perceptions of DPSI among physicians, nurses, and medical students.

### Questionnaire development and content

The questionnaire items were developed by reviewing previous studies related to DPSI [2, 4, 12, 13]. In particular, the questionnaire was developed to be consistent with the questionnaire items used to survey perception regarding DPSI among the general public [12, 13]. The members of the research team possessed experience conducting several studies in the field of patient safety, and questionnaire items that could comprehensively survey perceptions of DPSI were developed from the perspective of medical professionals—two physicians and three nurses. In

addition, the questionnaire was evaluated in a cognitive debriefing interview with 2 physicians and was revised according to their feedback.

The entire questionnaire survey consisted of the following: 1) assessment of level of knowledge of patient-safety-related terminology; 2) perception of DPSI under various conditions; 3) opinions of each component of DPSI in hypothetical cases; 4) opinions of methods for facilitating DPSI; and 5) socio-demographic questions. The present study focuses on analyzing the results on perception regarding DPSI under various conditions (#2 above) and on methods for facilitating DPSI (#4 above). More specifically, for #2, opinions about DPSI were gathered according to the seriousness of medical error, related situation, effects of DPSI, and barriers to DPSI. For #4, perceptions about enhancement of ethical consciousness, DPSI education and guidelines, and apology law, which prohibit certain statements or expressions related to DPSI from being admissible in a lawsuit, were gathered. For socio-demographic questions, information about the participants' sex, age group, and career stage (time since obtaining medical license) were collected.

## Administration of the survey and participants

An anonymous online questionnaire survey was conducted for about five months from October 2018 to February 2019. Any physician could participate in the survey and no specific exclusion criteria were set for the survey. This study was the first to examine the perception of DPSI among Korean physicians, so the sample size was not specifically set. The survey allowed as many participants as possible to participate in the survey, and ended by referring to the sample sizes from other similar studies [4].

Prior to the survey, the participants were presented with a definition of DPSI based on the assumption that they might not be familiar with DPSI terminology. To encourage participation by physicians, the target population, promotional messages were posted in various online physician communities (e.g. KakaoTalk group chat), and participants were also encouraged to promote the questionnaire survey to others to recruit more participants. The participants were given a small token of appreciation for their participation—a coffee voucher with monetary value of approximately 9,000 won (7 USD). The survey was set up such that each participant clicked through a unique link to complete the questionnaire, to prevent the same participant completing more than one questionnaire. Participants were required to complete the questionnaire in one sitting.

## Analysis

Descriptive analysis was performed to identify the response characteristics for each questionnaire item and the socio-demographic characteristics of the participants. Physicians' perceptions regarding DPSI as identified through the present study were compared to general public's perceptions of DPSI identified in previous studies [13]. The chi-squared test was performed to check for statistically significant differences in perceptions regarding DPSI between the two groups. All statistical analyses were performed using Stata/SE13.1 (StataCorp, Texas, TX), and $P<0.05$ was determined to be statistically significant.

## Ethics statement

This study was approved by the Institutional Review Board of the University of Ulsan Hospital (IRB Number: 2018-07-003). Prior to enrollment, we explained the objectives and processes of this study to the participants and obtained informed consent online from them. Only those who agreed to participate in the study conducted the survey.

## Results

### Socio-demographic characteristics

A total of 1,389 physicians participated in the survey, of which 910 (65.5%) completed the survey. The majority of the participants who completed the survey were males (n = 688, 75.6%) aged 30–39 years (n = 748, 82.2%). Regarding time since obtaining medical license, 5–9 years was the most common response (n = 614, 67.5%). More detailed socio-demographic characteristics of the participants are given in Table 1.

### DPSI according to the seriousness of medical error

A higher percentage of participants responded that harm caused by medical error should be disclosed to patients and/or their (family or other non-professional) caregivers when the harm was more severe (Table 2). For example, 94.9% of participants agreed that major errors should be disclosed to patients and/or their caregivers, whereas only 39.8% agreed that even near-miss incidents, which did not cause any harm to the patient, should be disclosed to patients and/or their caregivers. In contrast, 93.3% of the general public responded that even near-miss incidents should be disclosed to patients and/or their caregivers, a statistically significant difference.

### DPSI according to related situation

The participating physicians showed the highest level of agreement (87.4%) with the item "The better the previous physician–patient relationship, the more DPSI will be performed" (Table 3). Compared to the general public, there was a statistically significant difference in the level of agreement, but the difference was also smallest (5.3%) among the five items. On the other hand, the participating physicians showed the lowest level of agreement (64.4%) with the item "DPSI should be performed even if a physician thinks that patients and their caregivers have nothing to gain by having patient safety incidents acknowledged." Compared to the general public, there was again a statistically significant difference in the level of agreement, and the difference was largest (difference of 24.7%) among the five items.

### Perception regarding the effects of DPSI

Among the six known effects of DPSI presented, the participating physicians showed the highest level of agreement (89.6%) with the item "DPSI will lead physicians to pay more attention

**Table 1. Socio-demographic characteristics of survey participants.**

| Variable | | N | % |
|---|---|---|---|
| Age group | 19–29 | 140 | 15.4 |
| | 30–39 | 748 | 82.2 |
| | 40–49 | 10 | 1.1 |
| | ≥ 50 | 12 | 1.3 |
| Sex | Male | 688 | 75.6 |
| | Female | 222 | 24.4 |
| Career stage (time since obtaining medical license) | 0–4 | 208 | 22.8 |
| | 5–9 | 614 | 67.5 |
| | 10–19 | 77 | 8.5 |
| | ≥ 20 | 11 | 1.2 |
| Total | | 910 | 100.0 |

**Table 2. Perceptions of DPSI according to the level of harm resulting from medical errors.**

|  | Physician | | General Public | | P |
|---|---|---|---|---|---|
|  | Agree N (%) | Disagree N (%) | Agree N (%) | Disagree N (%) |  |
| Major errors should be disclosed to patients or their caregivers. | 864 (94.9) | 46 (5.1) | 699 (99.9) | 1 (0.1) | <0.001 |
| Minor errors should be disclosed to patients or their caregivers. | 714 (78.5) | 196 (21.5) | 685 (97.9) | 15 (2.1) | <0.001 |
| Near misses should be disclosed to patients or their caregivers. | 362 (39.8) | 548 (60.2) | 652 (93.3) | 47 (6.7) | <0.001 |

to patient safety in the future" (Table 4). This item showed the smallest difference in agreement between physicians and the general public (7.0%). On the other hand, the participating physicians showed the lowest level of agreement (62.4%) with the item "DPSI will lessen feelings of guilt for a physician." The item that showed the largest difference in level of agreement between physicians and general public was "DPSI will make patients and their caregivers trust the physician more," with agreement of 70.1% among the participating physicians and 94.1% among the general public (P<0.001).

## Perception regarding the barriers of DPSI

Among the six barriers to DPSI presented, the participating physicians showed the highest level of agreement (75.9%) with the item "It is unreasonable to demand DPSI in only the medical field, and disclosure is not actively conducted in other fields" (Table 5). This was also the item with the largest difference between physicians and the general public, with only 40.2% of the general public agreeing (P<0.001). On the other hand, the item "A physician who performs DPSI is less competent" showed the lowest level of agreement, with only 10.9% of the participating physicians agreeing. The level of agreement with this item among the general public was only 17.7%, showing the smallest difference between physicians and the general public among the items for barriers to DPSI (P<0.001). Meanwhile, the majority of both physicians and general public agreed with the item "DPSI will increase the incidence of medical lawsuits."

## Perception regarding method for facilitating DPSI

Among methods for facilitating DPSI, the participating physicians showed high level of agreement for the items "A guideline for DPSI is needed" (91.2%) and "Manpower to support DPSI in hospitals is required" (89.1%) (Table 6). Moreover, 79.3% of the participating physicians

**Table 3. Attitudes toward DPSI according to various scenarios in patient safety incidents.**

|  | Physician | | General Public | | P |
|---|---|---|---|---|---|
|  | Agree N (%) | Disagree N (%) | Agree N (%) | Disagree N (%) |  |
| DPSI should be performed even if a physician thinks that patients and their caregivers would not be able to understand what the physician said. | 776 (85.3) | 134 (14.7) | 694 (99.3) | 5 (0.7) | <0.001 |
| DPSI should be performed even if a physician thinks that patients and their caregivers would not want to know patient safety incidents. | 687 (75.5) | 223 (24.5) | 658 (94.0) | 42 (6.0) | <0.001 |
| DPSI should be performed even if a physician thinks that patients and their caregivers could not know whether patient safety incidents occurred without being told. | 707 (77.7) | 203 (22.3) | 670 (95.7) | 30 (4.3) | <0.001 |
| DPSI should be performed even if a physician thinks that patients and their caregivers have nothing to gain by having patient safety incidents acknowledged. | 586 (64.4) | 324(35.6) | 623(89.1) | 76(10.9) | <0.001 |
| The better the previous physician–patient relationship, the more DPSI will be performed. | 795 (87.4) | 115(12.6) | 649(92.7) | 51(7.3) | <0.001 |

**Table 4. Opinions on the effects of DPSI.**

| | Physician | | General Public | | P |
|---|---|---|---|---|---|
| | Agree N (%) | Disagree N (%) | Agree N (%) | Disagree N (%) | |
| DPSI will make patients and their caregivers trust the physician more. | 638 (70.1) | 272 (29.9) | 658 (94.1) | 41 (5.9) | <0.001 |
| I am more likely to recommend a physician who performs DPSI. | 652 (71.6) | 258 (28.4) | 597 (85.4) | 102 (14.6) | <0.001 |
| I will revisit a physician who performs DPSI. | 660 (72.5) | 250 (27.5) | 615 (88.0) | 84 (12.0) | <0.001 |
| A physician who performs DPSI will offer better medical services. | 630 (69.2) | 280 (30.8) | 623 (89.3) | 75 (10.7) | <0.001 |
| DPSI will lead physicians to pay more attention to patient safety in the future. | 815 (89.6) | 95 (10.4) | 675 (96.6) | 24 (3.4) | <0.001 |
| DPSI will lessen feelings of guilt for a physician. | 568 (62.4) | 342 (37.6) | 594 (85.1) | 104 (14.9) | <0.001 |

**Table 5. Perceptions of barriers to DPSI.**

| | Physician | | General Public | | P |
|---|---|---|---|---|---|
| | Agree N (%) | Disagree N (%) | Agree N (%) | Disagree N (%) | |
| DPSI will increase the incidence of medical lawsuits. | 610 (67.0) | 300 (33.0) | 399 (57.0) | 301 (43.0) | <0.001 |
| If DPSI is performed, a physician will lose his or her honor. | 333 (36.6) | 577 (63.4) | 239 (34.1) | 461 (65.9) | 0.308 |
| If DPSI is performed, the physician will be punished by his or her hospital. | 437 (48.0) | 473 (52.0) | 278 (39.8) | 421 (60.2) | 0.001 |
| A physician who performs DPSI is less competent. | 99 (10.9) | 811 (89.1) | 124 (17.7) | 575 (82.3) | <0.001 |
| If DPSI is performed, the physician will be criticized by his or her colleagues. | 316 (34.7) | 594 (65.3) | 291 (41.6) | 409 (58.4) | 0.005 |
| It is unreasonable to demand DPSI in only the medical field, and disclosure is not actively conducted in other fields. | 691 (75.9) | 219 (24.1) | 281 (40.2) | 418 (59.8) | <0.001 |

**Table 6. Opinions on methods for facilitating DPSI.**

| | Physician | | General Public | | P |
|---|---|---|---|---|---|
| | Agree N (%) | Disagree N (%) | Agree N (%) | Disagree N (%) | |
| It is necessary to strengthen the ethical mindset of physicians for DPSI. | 778 (85.5) | 132 (14.5) | 697 (99.6) | 3 (0.4) | <0.001 |
| A training course for DPSI is needed. | 794 (87.3) | 116 (12.7) | 682 (97.4) | 18 (2.6) | <0.001 |
| Manpower to support DPSI in hospitals is required. | 811 (89.1) | 99 (10.9) | 666 (95.3) | 33 (4.7) | <0.001 |
| A guideline for DPSI is needed. | 830 (91.2) | 80 (8.8) | 681 (97.3) | 19 (2.7) | <0.001 |
| If apology law is enacted, physicians will perform more DPSI. | 722 (79.3) | 188 (20.7) | 660 (94.3) | 40 (5.7) | <0.001 |
| Apology law will limit patients' ability to prove physicians' negligence. | 301 (33.1) | 609 (66.9) | 558 (79.7) | 142 (20.3) | <0.001 |
| I support the introduction of apology law. | 659 (72.4) | 251 (27.6) | 668 (95.4) | 32 (4.6) | <0.001 |
| I support the introduction of mandatory DPSI by law. | 349 (38.4) | 561 (61.6) | 634 (90.6) | 66 (9.4) | <0.001 |

agreed that "If an apology law is enacted, physicians will perform more DPSI," and 72.4% agreed that "I support the introduction of an apology law." However, only 33.1% of the participating physicians agreed that "An apology law will limit patients' ability to prove physicians' negligence," whereas 79.7% of the general public agreed with this item, showing a large difference in perception between the two groups (P<0.001). The item "I support the introduction of mandatory DPSI by law" also showed a large difference between the participating physicians (38.4%) and the general public (90.6%; P<0.001).

## Discussion

The present study investigated the opinions and perceptions of physicians in Korea regarding DPSI under various situations or conditions and methods for facilitating DPSI through an anonymous online questionnaire survey. Moreover, the results were compared to the general public's perceptions regarding DPSI, identified using the same questionnaire items, to examine differences between the two groups. Based on the findings, it was confirmed that, for the most part, physicians in Korea have positive perception of DPSI, but also that when compared to the absolute support for DPSI exhibited by the general public, physicians showed relatively low agreement on the need for and effectiveness of DPSI, especially in given cases. In particular, a low percentage of physicians believed that even near-miss errors should be disclosed, and they showed the lowest level of agreement among the various effects of DPSI that "DPSI will lessen feelings of guilt for a physician" (62.4%), while a majority of the physicians also showed concerns about increased medical lawsuits due to DPSI. As demonstrated by their agreement with the statement that "I support the introduction of an apology law," the participants showed positive perception, for the most part, of various methods for facilitating DPSI, despite also showing negative perceptions of legally mandating DPSI.

The most significant aspects of the present study are that it investigated current perceptions of DPSI among physicians in a non-Western country. Because most previous studies that quantitatively examined perceptions of DPSI among physicians were conducted in Western countries prior to 2010, they have the limitation of not reflecting the latest status of DPSI perceptions among physicians, in Western countries or globally [6–8, 10, 11]. Moreover, there are very few studies that comprehensively investigate perception of barriers to and facilitating methods for DPSI, or of how to perform DPSI under various situations and conditions, as the present study did. Monitoring changes in perception of DPSI among physicians by conducting regularly scheduled questionnaire surveys, using or making reference to the items used in the present study, should help establish measures to effectively implement DPSI in clinical practice [10].

Among relevant previous studies, the latest was by Iezzoni et al. published in 2012, in which open and honest communication with patients was investigated through a questionnaire survey of 1,891 physicians [9]. Although direct comparison between that study and the present study has limitations due to differences in the questionnaire items, methodologies, and characteristics of study participants, the findings of the present study show little difference in perception of DPSI when compared to that survey, conducted in 2009. For example, according to Iezzoni et al., only 65.9% of participants completely agreed that physicians should "disclose all significant medical errors to affected patients," much as only 78.5% of the participants in the present study agreed that "minor errors should be disclosed to patients or their caregivers." However, it is necessary to conduct follow-up studies with the same method and items to increase comparability.

A noteworthy finding in the present study was that there is a difference in perception regarding DPSI between physicians and the general public. First, regarding perceptions of the need for DPSI according to the level of harm caused by medical error, the results showed a large difference when near-miss errors occur. While 93.3% of the general public responded that even near-miss errors should be disclosed to patients and/or their caregivers [13], only 39.8% of the participating physicians agreed. This finding was consistent with the results of previous studies [14–16]. The present study did not identify specific reasons for the responses given, but it is probable that the physicians were concerned that disclosing near-miss errors could possibly cause patients and their caregivers to lose trust in medical staff [2]. However, considering that 89.6% of physicians agreed with the item "DPSI will lead physicians to pay more attention to patient safety in the future," it may be necessary to promote the need to disclose near-miss errors in order to

increase medical professionals' recognition of various patient safety issues, including DPSI. This is because disclosing near-miss incidents, which lead to no specific harm to patients, does not create compensation issues, and there is evidence that DPSI can increase the level of trust that patients and their caregivers have in medical staff [4].

The physicians participating in the present study mostly recognized the various known effects of DPSI. However, while a majority of the participating physicians agreed with the item "DPSI will lessen feelings of guilt for a physician," the level of agreement for that item was lower than that for other effects. Physicians involved in patient safety incidents are known to experience emotional suffering, and as such, they are often referred to as the "second victims" of such incidents [17]. A previous study conducted in Korea confirmed that physicians who experienced a patient safety incident felt embarrassment and fear as well as a great sense of regret and guilt towards the patients and their caregivers [18]. According a systematic literature review, DPSI is known to reduce physicians' guilty feeling [4], but consideration should be given to the fact that the level of decrease may not be as large as researchers think. Therefore, there is the need to establish a system that can help at an institutional level to not only implement DPSI policies within an institution, but also organize counseling services to provide support to second victims [19].

With respect to the survey results on perception of barriers to DPSI, the physicians participating in the present study showed the highest level of agreement (75.9%) to the item "It is unreasonable to demand DPSI in only the medical field, and disclosure is not actively conducted in other fields." Previous studies have investigated perception of various barriers to DPSI, such as fear about medical lawsuits and disciplinary actions or loss of trust from patients and colleagues [4], but almost no questionnaire surveys have asked questions about the possibility of uneven burden of demanding DPSI in the medical field but not, or in relation to, other fields. The findings of the present study are believed to have been influenced by dissatisfaction physicians feel toward government policies and the burden of societal demand for high ethical standards. Moreover, the present study also reconfirmed that fear of medical lawsuits is a major barrier to DPSI. According to Iezzoni et al., approximately 80% of physicians surveyed completely agreed with the item about being reluctant to disclose all mistakes to patients due to fear of lawsuits [9]. It is believed that a similar sentiment motivates the 67.0% of the participants in the present study who agreed with the item "DPSI will increase the incidence of medical lawsuits." However, according a previous study in Korea that investigated the effects of DPSI among the general public using hypothetical cases, DPSI lowered the intention to file medical lawsuits and lowered criminal prosecution [12]. It is necessary to promote more awareness of these effects, perhaps by establishing a system for medical professionals who have experienced success with DPSI to share their experiences.

There is a need more generally to explore methods of facilitating DPSI in clinical practice. The physicians participating in the present study mostly agreed with various methods for facilitating DPSI. It is necessary for the government to provide policy support by establishing guidelines to regulate how to perform DPSI based on the approaches physicians support and their expertise in DPSI, developing and applying educational programming on this subject; and creating support teams within medical institutions. It is necessary to reference DPSI guidelines such as those developed in Canada and Australia to develop national Korean DPSI guidelines that reflect the reality of clinical practice in Korea and consider the preferences of Korean medical professionals, and to distribute such guidelines to medical institutions and implement policies recommending the use of such guidelines [20, 21].

Among legislative methods for facilitating DPSI, confirming support among physicians for an apology law is another significant finding of the present study. From the perspective of medical professionals, there is no particular reason to oppose an apology law, which would stipulate that

sympathy, regret, or apology expressed in the course of performing DPSI would not be recognized as admission of legal responsibility during a civil medical lawsuit [22], but there has been no effort to determine the level of (dis)approval of an apology law among medical professionals, including physicians. It may be necessary to begin legal review of whether an apology law could be implemented in Korea, given the support for an apology law among the general public demonstrated in a previous study [13] and the support among physicians confirmed in the present study [1, 23]. However, considering that there is a large opinion gap between physicians and the general public on legally mandating DPSI, some controversy may be expected if this is done.

The limitations of the present study include the potentially limited representativeness of the physicians who participated. The study attempted to overcome any representativeness issue by having as many physicians as possible participate; however, due to the nature of the anonymous online questionnaire survey method, this representativeness issue could not be completed resolved. Nevertheless, using this type of questionnaire survey may be better for obtaining honest opinions from physicians about their views on DPSI, such as on ethical issues. It may be necessary to conduct future studies with various physician groups using a questionnaire similar to the one used in the present study and compare the findings between studies. Moreover, conducting similar questionnaire surveys with other medical professionals besides physicians, such as nurses and pharmacists, and comparing the results may be meaningful as well.

## Conclusions

The present study investigated the perceptions of DPSI among physicians in Korea from various perspectives, including effects, barriers, and facilitating methods. Based on the findings, it was determined that a gap exists between physicians and general public in perception of DPSI. In order to reduce the gap in perception of DPSI between the two groups, various strategies such as education and promotion of DPSI for physicians as well as the general public are required. Moreover, the questionnaire survey items used in the present study should be generalizable and helpful to comprehensively examine physicians' perceptions of DPSI in different countries and medical institutions that have plans to implement or investigate DPSI. Regular questionnaire surveys assessing DPSI attitudes among various relevant populations will be helpful to promote understanding of the importance and benefits of DPSI.

## Supporting information

**S1 File. Questionnaire (English version).**
(DOCX)

**S2 File. Questionnaire (Korean version).**
(DOCX)

## Author Contributions

**Conceptualization:** Jeehee Pyo, Eun Young Choi, Won Lee, Seung Gyeong Jang, Minsu Ock, Sang-Il Lee.

**Data curation:** Jeehee Pyo, Young-Kwon Park, Minsu Ock.

**Formal analysis:** Jeehee Pyo, Young-Kwon Park, Minsu Ock.

**Methodology:** Jeehee Pyo, Minsu Ock.

**Validation:** Jeehee Pyo, Eun Young Choi, Won Lee, Seung Gyeong Jang, Young-Kwon Park, Minsu Ock, Sang-Il Lee.

**Writing – original draft:** Jeehee Pyo, Minsu Ock.

**Writing – review & editing:** Jeehee Pyo, Eun Young Choi, Won Lee, Seung Gyeong Jang, Young-Kwon Park, Minsu Ock, Sang-Il Lee.

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
