## [Decision Letter · Decision Letter 0]

17 Aug 2020

PONE-D-20-17785

Korean physicians’ perceptions regarding disclosure of patient safety incidents: A cross-sectional study

PLOS ONE

Dear Dr. Ock,

Thank you for submitting your manuscript to PLOS ONE. After careful consideration, we feel that it has merit but does not fully meet PLOS ONE’s publication criteria as it currently stands. Therefore, we invite you to submit a revised version of the manuscript that addresses the points raised during the review process.

We look forward to receiving your revised manuscript.

Kind regards,

Tim Schultz

Academic Editor

PLOS ONE

Journal Requirements:

2. In your Methods section, please provide additional information about the participant recruitment method and the demographic details of your participants. Please ensure you have provided sufficient details to replicate the analyses such as: a) the recruitment date range (month and year), b) a description of any inclusion/exclusion criteria that were applied to participant recruitment, c) a table of relevant demographic details, d) a statement as to whether your sample can be considered representative of a larger population, e) a description of how participants were recruited, and f) descriptions of where participants were recruited and where the research took place.

3. Please include additional information regarding the survey or questionnaire used in the study and ensure that you have provided sufficient details that others could replicate the analyses. For instance, if you developed a questionnaire as part of this study and it is not under a copyright more restrictive than CC-BY, please include a copy, in both the original language and English, as Supporting Information. Moreover, please include more details on how the questionnaire was pre-tested, and whether it was validated.

Additional Editor Comments (if provided):

Two reviewers' comments are provided below. Please respond to reviewer 2's statement that "the presented results in this study is already presented in the previous study and it is hard to find novel finding". This response may or may not require revisions to the manuscript.

Please also respond to reviewer 1's statement that the manuscript mainly addresses differences between consumers' and doctors' perspectives, and "it would be good to consider points to be discussed on the side of the general public for improvement of the areas perceived as barriers by physicians and the general public". Additionally, please respond to the reviewers comments about limitations.

From an editorial perspective, I would like to see a completed STROBE checklist, and the manuscript should be revised to address (at least) items: #5 (dates of administation of survey, which is important when considered that comparisons are being made to the 2015 data collected on consumers), #6 inclusion criteria, #10 sample size, #13 participants, and #21 generalisability.

There is a typo on line 149 (setting for sitting).

Lastly, can you explain why medical specialty was not recorded?

Reviewers' comments:

Reviewer's Responses to Questions

**Comments to the Author**

1. Is the manuscript technically sound, and do the data support the conclusions?

Reviewer #1: Yes

Reviewer #2: Yes

2. Has the statistical analysis been performed appropriately and rigorously? 

Reviewer #1: Yes

Reviewer #2: Yes

3. Have the authors made all data underlying the findings in their manuscript fully available?

Reviewer #1: Yes

Reviewer #2: Yes

4. Is the manuscript presented in an intelligible fashion and written in standard English?

Reviewer #1: Yes

Reviewer #2: Yes

5. Review Comments to the Author

Reviewer #1: Thank you for the opportunity to review this article, titled “Korean physicians’ perceptions regarding disclosure of patient safety incidents: A cross-sectional study.”

My suggestions for revision are:

Although the title of the paper says “Korean physicians’ perceptions regarding disclosure of patient safety incidents: A cross-sectional study,” the author emphasized the gap between physicians’ perceptions and the general public’s perceptions from the introduction to the conclusion of the paper. Although the difference between physicians' perceptions and general public's perceptions was well presented through comparison, the discussion seems to be mainly concentrated on improvements and suggestions on the side of physicians. While physicians’ perceptions should be changed and legal and institutional support for change is necessary, the approach on the side of the general public also seems to be important, but there seems to be a lack of discussion of it. In particular, although measures to narrow the gap between physicians' perceptions and general public's perceptions of barriers to DPSI are important, it would be good to consider points to be discussed on the side of the general public for improvement of the areas perceived as barriers by physicians and the general public.

In addition, the researcher emphasized the necessity of repeated studies with expanded study subjects (applying various countries, institutions, occupations, etc.). If the author has a concern about areas that could not be measured or areas that should be corrected or supplemented in the questionnaire applied in this study, it may be good to add them to the limitations.

Thank you for your writing a good paper.

Reviewer #2: Error disclosure is very important issue for patient safety and the authors surveyed Korean physicians’ perceptions regarding error disclosure. However, a few critics need to be addressed for this study.

This study presents Korean physicians’ perceptions regarding error disclosure according to level of harm from medical errors, various situations, barriers to error disclosure and methods for facilitating it, etc.

But the presented results in this study is already presented in the previous study and it is hard to find novel finding.

6. PLOS authors have the option to publish the peer review history of their article (what does this mean?). If published, this will include your full peer review and any attached files.

Reviewer #1: No

Reviewer #2: No

---

## [Author Response · Author response to Decision Letter 0]

11 Sep 2020

Journal Requirements:

Response: We would like to thank you for giving us an opportunity to revise our manuscript. Our responses follow.

Response: We rechecked our manuscript to meet the PLOS ONE's style requirements and revised it accordingly.

2. In your Methods section, please provide additional information about the participant recruitment method and the demographic details of your participants. Please ensure you have provided sufficient details to replicate the analyses such as: a) the recruitment date range (month and year), b) a description of any inclusion/exclusion criteria that were applied to participant recruitment, c) a table of relevant demographic details, d) a statement as to whether your sample can be considered representative of a larger population, e) a description of how participants were recruited, and f) descriptions of where participants were recruited and where the research took place.

Response: As you suggested, we revised the Methods section for providing more details of the survey, including the recruitment date range and inclusion and exclusion criteria for the survey (Line 141~). As previously stated, the survey was conducted online, and promotional messages were posted in various online physician communities to encourage the participation of physicians. Also, participants were encouraged to promote the questionnaire survey to other physicians for further recruitment. In order to obtain an honest survey response, we constructed the survey that minimizes the collection of socio-demographic characteristics of participants. Therefore, we collected only the following socio-demographic characteristics of participants: age group, sex, and time since obtaining a medical license. The study attempted to overcome any representativeness issue by having as many physicians as possible to participate in the study. However, due to the nature of the anonymous online questionnaire survey method, this representativeness issue could not be completed resolved. We have already acknowledged this point as a limitation.

3. Please include additional information regarding the survey or questionnaire used in the study and ensure that you have provided sufficient details that others could replicate the analyses. For instance, if you developed a questionnaire as part of this study and it is not under a copyright more restrictive than CC-BY, please include a copy, in both the original language and English, as Supporting Information. Moreover, please include more details on how the questionnaire was pre-tested, and whether it was validated.

Response: We did not include the questionnaire as we expected the readers to speculate the content of the questionnaire through the tables. However, as you suggested, we have attached an English and Korean version questionnaire as Supplementing Information. As described in the Method section, the questionnaire was developed to be consistent with the questionnaire items used to survey perceptions regarding DPSI among the general public. The questionnaire was evaluated in a cognitive debriefing interview with two physicians and was revised according to their feedback (Line 126~).

Additional Editor Comments (if provided):

Two reviewers' comments are provided below. Please respond to reviewer 2's statement that "the presented results in this study is already presented in the previous study and it is hard to find novel finding". This response may or may not require revisions to the manuscript.

Please also respond to reviewer 1's statement that the manuscript mainly addresses differences between consumers' and doctors' perspectives, and "it would be good to consider points to be discussed on the side of the general public for improvement of the areas perceived as barriers by physicians and the general public". Additionally, please respond to the reviewers comments about limitations.

Response: We would like to thank reviewers for a careful and thorough reading of this manuscript and for the thoughtful comments and constructive suggestions, which help to improve the quality of this manuscript. Our responses can be found in the reviewers' comments below.

From an editorial perspective, I would like to see a completed STROBE checklist, and the manuscript should be revised to address (at least) items: #5 (dates of administation of survey, which is important when considered that comparisons are being made to the 2015 data collected on consumers), #6 inclusion criteria, #10 sample size, #13 participants, and #21 generalisability.

Response: As you suggested, we reviewed the STROBE checklist and revised the manuscript according to it. We added recruitment date range and inclusion and exclusion criteria for the survey, rationale of sample size, and more details of survey participants (Line 141~, Line 177~). As we acknowledged in the Discussion section, the limitations of the present study include the potentially limited representativeness of the physicians who participated in the study.

There is a typo on line 149 (setting for sitting).

Response: We corrected the typo (Line 155).

Lastly, can you explain why medical specialty was not recorded?

Response: In order to obtain an honest survey response, we constructed the survey that minimizes the collection of socio-demographic characteristics of participants. Therefore, we collected only the following socio-demographic characteristics of participants: age group, sex, and time since obtaining a medical license.

 

Reviewers' comments:

Reviewer's Responses to Questions

Comments to the Author

1. Is the manuscript technically sound, and do the data support the conclusions?

Reviewer #1: Yes

Reviewer #2: Yes

2. Has the statistical analysis been performed appropriately and rigorously? 

Reviewer #1: Yes

Reviewer #2: Yes

3. Have the authors made all data underlying the findings in their manuscript fully available?

Reviewer #1: Yes

Reviewer #2: Yes

4. Is the manuscript presented in an intelligible fashion and written in standard English?

Reviewer #1: Yes

Reviewer #2: Yes

5. Review Comments to the Author

Reviewer #1: Thank you for the opportunity to review this article, titled “Korean physicians’ perceptions regarding disclosure of patient safety incidents: A cross-sectional study.”

Response: We would like to thank you for the careful and thorough reading of this manuscript and for the thoughtful comments and constructive suggestions, which help to improve the quality of this manuscript. Our responses follow.

My suggestions for revision are:

Although the title of the paper says “Korean physicians’ perceptions regarding disclosure of patient safety incidents: A cross-sectional study,” the author emphasized the gap between physicians’ perceptions and the general public’s perceptions from the introduction to the conclusion of the paper. Although the difference between physicians' perceptions and general public's perceptions was well presented through comparison, the discussion seems to be mainly concentrated on improvements and suggestions on the side of physicians. While physicians’ perceptions should be changed and legal and institutional support for change is necessary, the approach on the side of the general public also seems to be important, but there seems to be a lack of discussion of it. In particular, although measures to narrow the gap between physicians' perceptions and general public's perceptions of barriers to DPSI are important, it would be good to consider points to be discussed on the side of the general public for improvement of the areas perceived as barriers by physicians and the general public.

Response: Thank you for your helpful feedback. As you mentioned, this study looked at the physicians' perception of DPSI and compared it with the results of the general public. Since the results of the general public’s perception were mainly dealt with in the previous article (Ock M, Choi EY, Jo MW, Lee SI. General public's attitudes toward disclosure of patient safety incidents in Korea: results of disclosure of patient safety incidents survey I. J Patient Saf 2017), we mainly focused on the physicians' perception in this manuscript. Based on the findings from this study, it was confirmed that, for the most part, physicians in Korea have positive perceptions of DPSI, but also that when compared to the complete support for DPSI exhibited by the general public, they showed relatively low agreement on the need for and effectiveness of DPSI. In the Discussion section, we reviewed the items that demonstrated significant differences in perceptions between the two groups in detail: disclosure of near-miss, apology law, and DPSI law. The conclusion emphasized the need for proper DPSI awareness by physicians as well as the general public to reduce the differences in perceptions, as you suggested (Line 352~).

In addition, the researcher emphasized the necessity of repeated studies with expanded study subjects (applying various countries, institutions, occupations, etc.). If the author has a concern about areas that could not be measured or areas that should be corrected or supplemented in the questionnaire applied in this study, it may be good to add them to the limitations.

Thank you for your writing a good paper.

Response: One of the strengths of this study is that the survey items could comprehensively evaluate the perceptions of DPSI. We hope that our follow-up study will improve DPSI by conducting similar surveys and comparing them to one another. Therefore, we have attached an English and Korean version of the questionnaire as Supplementing Information for further research.

Reviewer #2: Error disclosure is very important issue for patient safety and the authors surveyed Korean physicians’ perceptions regarding error disclosure. However, a few critics need to be addressed for this study.

Response: We would like to thank you for the careful and thorough reading of this manuscript and for the thoughtful comments and constructive suggestions, which help to improve the quality of this manuscript. Our responses follow.

This study presents Korean physicians’ perceptions regarding error disclosure according to level of harm from medical errors, various situations, barriers to error disclosure and methods for facilitating it, etc. But the presented results in this study is already presented in the previous study and it is hard to find novel finding.

Response: As we described in the Introduction section, while there are previous studies quantitatively examining perceptions of physicians regarding DPSI, most of these do not reflect the latest phenomenon due to their dates of research. Moreover, as most of these studies were conducted in Western countries, possible cultural differences in perceptions regarding DPSI cannot be dismissed. Also, few studies quantitatively examined the differences in perceptions of DPSI between physicians and the general public, including patients. Therefore, we conducted a questionnaire survey among physicians in Korea to investigate their perceptions of DPSI from various perspectives, including its effects, barriers to it, and methods facilitating it. These points are expected to be the strengths of our study.

6. PLOS authors have the option to publish the peer review history of their article (what does this mean?). If published, this will include your full peer review and any attached files.

Do you want your identity to be public for this peer review? For information about this choice, including consent withdrawal, please see our Privacy Policy.

Reviewer #1: No

Reviewer #2: No

---

## [Editor Report · Decision Letter 1]

25 Sep 2020

Korean physicians’ perceptions regarding disclosure of patient safety incidents: A cross-sectional study

PONE-D-20-17785R1

Dear Dr. Ock,

We’re pleased to inform you that your manuscript has been judged scientifically suitable for publication and will be formally accepted for publication once it meets all outstanding technical requirements.

Kind regards,

Tim Schultz

Academic Editor

PLOS ONE

Additional Editor Comments (optional):

Thank-you for submitting R1 of this manuscript, which addresses the reviewer feedback.

I am happy to accept the paper subject to one additional clarification, which relates earlier requested revisions around describing the participants and how they were recruited/selected (eg 6(a) of STROBE).

The description of the recruitment strategy line 149-50 is a little vague "various online physician communities", can you please provide additional information about this to enhance repeatability of your study.
---

## [Editor Report · Acceptance letter]

30 Sep 2020

PONE-D-20-17785R1 

Korean physicians’ perceptions regarding disclosure of patient safety incidents: A cross-sectional study 

Dear Dr. Ock:

I'm pleased to inform you that your manuscript has been deemed suitable for publication in PLOS ONE. Congratulations! Your manuscript is now with our production department. 

Kind regards, 

on behalf of

Dr. Tim Schultz 

Academic Editor

PLOS ONE